# Integrated gene network analysis and experimental validation identify key hub genes in potato response to Potato Virus Y infection

Roya Karimipour[1], Davoud Koolivand [1]*, Abozar Ghorbani [2]*, Masoud Naderpour [3], Mahsa Rostami[2]

1 Department of Plant Protection, Faculty of Agriculture, University of Zanjan, Zanjan, Iran, 2 Nuclear Agriculture Research School, Nuclear Science and Technology Research Institute (NSTRI), Karaj, Iran, 3 Seed and Plant Certification and Registration Research Institute (SPCRI), Agricultural Research, Education and Extension Organization (AREEO), Karaj, Iran

* koolivand@znu.ac.ir (DK), abghorbany@aeoi.org.ir (AG)

## Abstract

Potato (*Solanum tuberosum*) is a staple food crop that supports global food security, ranking as the world's third most important food crop after rice and wheat in terms of human consumption, and it is threatened by Potato virus Y (PVY), which causes severe yield losses. This study integrates bioinformatics analysis and experimental approaches to elucidate molecular defense mechanisms against PVY infection. Using transcriptomic data from PVY-infected potato plants, we constructed protein-protein interaction (PPI) networks and identified hub genes central to defense responses. The qPCR validation showed that three hub genes (*NAD1*, *NAD2*, *NAD3*) were upregulated in resistant Sante plants but downregulated in susceptible Agria. Among these, *NAD2* showed a striking 5.58-fold increase in Sante, highlighting its critical role in stress signaling and antiviral defense. Network analysis revealed interactions with microRNAs (miRNAs), including stu-miR8015-5p and stu-miR396-5p, suggesting complex regulatory networks. Codon usage bias analysis highlighted adaptive codon preferences optimized for translational efficiency, supporting potential strategies like codon deoptimization to impair viral fitness. Promoter motif analysis identified stress-responsive *cis*-regulatory elements linked to abscisic acid signaling, critical for antiviral responses. This comprehensive study establishes a framework for targeting hub genes and miRNAs to engineer PVY-resistant cultivars, thereby offering a sustainable solution.

## Introduction

Potato (*Solanum tuberosum*) is a globally important crop that serves as a cornerstone of food security and agricultural economies. Despite its importance, potato production and quality are often compromised by biotic stresses, with viral infections

**Data availability statement:** All relevant data are within the manuscript and its Supporting Information files.

**Funding:** The author(s) received no specific funding for this work.

**Competing interests:** The authors have declared that no competing interests exist.

posing some of the most significant challenges. Among these, Potato virus Y (PVY) stands out as one of the most damaging pathogens, causing symptoms such as mosaic patterns, leaf necrosis, and a reduction in tuber yield and quality. PVY, a member of the Potyviridae family, is primarily transmitted by aphids in a non-persistent manner, with additional modes of transmission including infected tubers and mechanical means via contaminated tools or handling. The widespread impact of PVY infection, including Potato Tuber Necrotic Ringspot Disease (PTNRD) caused by aggressive strains (PVYNTN, PVYN), results in significant economic losses through both yield reduction and tuber quality degradation. This underscores the critical need to understand molecular defense mechanisms and develop effective control strategies [1].

Understanding the molecular mechanisms of PVY resistance is imperative for the development of effective strategies to enhance resistance. For instance, transcriptomic studies of potato leaves infected by PVY have revealed substantial transcriptional shifts, with processes such as photosynthesis, nucleosome assembly, and lipid transfer activity demonstrating differential regulation based on the compatibility between the host and the virus [2]. Moreover, weighted gene co-expression network analysis is a robust tool for identifying critical genes involved in plant responses to viral infections like PVY. This network-based strategy, particularly through the identification of hub genes, has been extensively utilized to elucidate the molecular pathways underlying resistance mechanisms. Using this approach, three hub genes were discovered to play pivotal roles in the defense response to PVY. These genes are integral to key biological processes, including chloroplast degradation, the regulation of defense-related proteins in infected cells, and the alkalization of the apoplast. Their identification underscores their essential role in preserving network connectivity and highlights their potential application as genetic markers for developing PVY-resistant potato cultivars [3].

Despite the substantial research conducted on biotic and abiotic stress and their interactions with host plants, a notable gap remains in the integration of computational network analyses with experimental validation to identify the critical regulatory components involved in plant defense responses. The resolution of this lacuna has the potential to significantly enhance our understanding of the molecular interactions that determine resistance or susceptibility to infection. While bioinformatics tools, such as co-expression network analysis, offer valuable predictions of hub genes and their functions, experimental validation remains crucial for confirming their specific roles in plant defense mechanisms. The potential of network analysis in enhancing crop resistance and refining plant breeding approaches is underscored by these insights [4].

In this study, an integrated gene network analysis was performed to identify critical hub genes associated with potato responses to PVY infection. Publicly available transcriptomic datasets were utilized to construct and analyze gene regulatory networks, allowing for the identification of hub genes with high centrality measures. To validate these findings, experimental real-time PCR analysis was conducted on three candidate hub genes, and their differential expression in response to PVY infection was

confirmed. The integration of computational and experimental approaches yielded a comprehensive framework for elucidating pivotal regulatory elements in potato defense against PVY.

## Materials and methods

### Plant materials and virus inoculation

The Seed and Plant Certification and Registration Institute (SPCRI), Karaj, Alborz, supplied the commercial potato cultivars Agria (susceptible) and Sante (resistant) [5], as well as virus inoculum in the form of a PVY-infected potted potato plant. The virus was mechanically inoculated onto four-week-old potato plants at the four-leaf stage under standard greenhouse conditions [6]. Following inoculation, the plants were maintained in the greenhouse for symptom development and subsequent analysis. The presence of PVY in infected plants was confirmed using an indirect double-antibody sandwich enzyme-linked immunosorbent assay (ELISA) with specific PVY recombinant antibody (polyclonal) obtained from the SPCRI, Karaj, Alborz. Statistical significance between infected and control groups was determined using Student's t-test ($P < 0.05$). Three replicates were analyzed per group.

### Collection of genes associated with PVY infection

In a previous study [7], genes exhibiting significant overexpression during PVY infection were identified through the use of transcriptome profiling of potato plants. For the present study, genes with a fold change greater than 2 or less than −2 were selected for network analysis (S1 File).

### Protein-protein interaction networks and hub gene analysis

Protein-protein interaction (PPI) networks for the selected genes (S1 File) were analyzed using the web-based STRING platform (version 10, http://string-db.org) with a minimum interaction score threshold of 0.150 (low confidence). The resulting PPI data were imported into Cytoscape (version 3.9.1) for further analysis. To identify hub genes within the network, the CytoHubba plugin (version 0.1), which is integrated into Cytoscape, was employed. Four computational algorithms, MCC, Degree, DMNC, and MNC, available in CytoHubba were utilized to rank and identify hub genes. For each algorithm, four proteins were selected as hub proteins. The identified hub genes and their interactions were then visualized as a subnetwork.

### Gene ontology and pathway enrichment analysis of differentially expressed genes

The Kyoto Encyclopedia of Genes and Genomes (KEGG) database was utilized to perform pathway enrichment analysis for the hub genes identified in the subnetwork. Additionally, gene ontology (GO) analysis, encompassing molecular function (MF), cellular component (CC), and biological process (BP) categories, was conducted using the web-based STRING platform.

### Network cluster analysis

The CytoCluster plugin (version 2.1.0) was used to identify clusters within the network nodes. For cluster analysis of the subnetwork, the protein complex identification algorithm (IPCA) was applied with a threshold of 10 [8]. Genes within each identified cluster were further analyzed using the STRING platform (version 10) to identify the KEGG pathways associated with these gene clusters.

### Promoter motif analysis of hub genes

The 1 kb upstream flanking regions (UFRs) of hub genes were retrieved from the Phytozome database (http://www.phytozome.net). Conserved motifs within these sequences were identified using the MEME Suite (version 5.4.1) (meme.nbcr.net/

meme/intro.html) (Bailey et al., 2009) with default parameters except for P-value and E-value thresholds, which were set to <0.01. To eliminate redundant motifs and identify known *cis*-regulatory elements (CREs), the Tomtom tool (version 5.4.1) (http://meme-suite.org/tools/tomtom) was used with thresholds of <0.01 for P-values and <0.1 for E-values, based on the JASPAR CORE 2022 database (Gupta et al., 2007). Additionally, the GOMo tool (http://meme-suite.org/tools/gomo) was used to predict potential functional roles for the identified motifs [9].

### Identification of potential microRNAs targeting hub genes

The potential miRNAs associated with the hub genes were identified using the psRNATarget server (http://plantgrn. noble.org/psRNATarget/). Potato miRNAs were downloaded from the miRbase database (https://www.mirbase.org/). The sequences of the hub genes were analyzed with default parameters, except for the maximum expectation value, which was set to 4. The results were then compared with 80 published miRNAs of *S. tuberosum* to identify relevant matches.

### Codon usage analysis of hub genes

To examine codon preference in the hub genes, the open reading frame (ORF) regions for each gene were identified using the Phytozome database (http://www.phytozome.net). ORF regions shorter than 100 bp or containing a termination codon were excluded from the analysis. Codon preference patterns were evaluated using indices such as the Codon Adaptation Index (CAI), Effective Number of Codons (ENC), GC content, GC content at the third codon position (GC3S), and Relative Synonymous Codon Usage (RSCU). Statistical analyses were conducted with R software (version 3.4.2, https://www.r-project.org) to explore relationships and potential correlations between these indices, codon usage patterns, and gene expression levels.

### Virus inoculation and hub gene expression analysis in potato cultivars

Six pots per potato cultivar were grown and maintained under standard greenhouse conditions. The experiment was set up in plastic pots containing a sterilized soil mixture and arranged in a completely randomized design. After four weeks of growth, virus inoculation was carried out using infected leaf tissue when the plants reached the four-leaf stage. PVY was introduced by mechanical inoculation, while uninoculated plants served as controls. Pots were monitored daily, watered as needed, and maintained at a greenhouse temperature of 25–30°C. Initial virus symptoms, such as leaf curling, were observed one week after inoculation. At the onset of symptoms, leaf samples were collected from both infected and control plants for RNA extraction. Stored leaf samples were crushed in liquid nitrogen using a mortar and pestle before RNA isolation. Total RNA was extracted using the RNA purification kit (DENA Zist Asia, Iran) according to the manufacturer's instructions. Total RNA quality was verified by 1.5% agarose gel electrophoresis, showing intact 18S and 28S ribosomal RNA bands with no visible degradation. The extracted RNA was treated with DNase (Thermo Scientific) for first-strand cDNA synthesis using the cDNA synthesis kit (Pishgam, Iran).

For gene expression analysis, real-time PCR was performed using primers designed for hub genes identified by network analysis. The primers for the target genes (*NAD1*, *NAD2*, and *NAD3*) were designed using the Primer3 software [10], based on three randomly selected hubs identified through network analysis. Also, the *EF1* gene was used as an internal control to normalize the expression levels of the target genes across the sample [11] (Table 1). The primers were ordered from BioMagic Gene Company. Real-time PCR was performed using the Mic Real-Time PCR Cycler (Bio Molecular Systems, Australia) and REALQ PLUS 2X MASTER MIX GREEN (Ampliqon) in a total reaction volume of 12.5 µL. Each reaction mixture contained 20 ng of cDNA and 10 µM of each primer. The qPCR cycling conditions included an initial denaturation at 94°C for 15 minutes, followed by 35 cycles of 95°C for 30 seconds, 57°C for 30 seconds, and 72°C for 30 seconds. Each run included no-template controls (NTC) with nuclease-free water replacing cDNA and no-reverse transcription controls (no-RT) to monitor genomic DNA contamination. Moreover, melt curve analysis (65–95°C with 0.5°C increments) confirmed single amplification products for all primer pairs.

**Table 1. Sequences of the primer sets used for quantitative real-time PCR.**

| Gene name | Primer sequence |
| --- | --- |
| EF1α F | GATGGTCAGACCCGTGAACA |
| EF1α R | CCTTGGAGTACTTCGGGGTG |
| NAD1F | CCAAGGAAGCAGCGCCTCTA |
| NAD1R | GCATTCGCGGCCTCACTAG |
| NAD2F | TGCAGTAACGCTGGTGTATTAGG |
| NAD2R | GGGACACCGGCACCAAAG |
| NAD3F | CCGGTGCGTCATCAGGAATC |
| NAD3R | TGCACGTTGAGGTCCGTTTG |

The experiment had three biological replicates and two technical replicates for each sample. Threshold cycle (Ct) values were recorded, and transcript abundance was analyzed in Microsoft Excel by normalizing Ct values to the EF1 gene. The relative expression of target genes was determined using the $2^{-\Delta\Delta CT}$ method [12]. A two-way ANOVA was performed to assess differences in gene expression between groups. Following a significant interaction effect ($P < 0.05$), Tukey's HSD post-hoc test was applied for pairwise comparisons.

## Result

### Virus infection of potato plants

In a published study [7], both healthy and PVY-infected potato plants underwent whole transcriptome sequencing with three biological replicates per group. In our experiments, infected potato plants were observed for typical PVY symptoms, including leaf curling, chlorosis and stunting, which became evident one week after inoculation under controlled greenhouse conditions. The presence of PVY in inoculated plants was confirmed by an ELISA test (Fig 1).

### Analysis of PPI networks and identification of hub genes

In this study, a systems biology approach was applied to analyze 173 genes derived from the differentially expressed genes (DEGs) identified by a published study [7]. We used both up-and down-regulated genes from PVY-infected potato plants and constructed PPI networks using the STRING database and Cytoscape (Fig 2). STRING (http://string-db.org) is a well-established database that compiles both predicted and experimentally confirmed protein interactions [13] and is often used in the methods section for such analyses. Cytoscape, on the other hand, is an open-source platform designed to visualize molecular interaction networks and integrate related data [14]. Integrated network analysis identified ATP synthase subunits and mitochondrial complex I components as top-ranked hub genes, underscoring their pivotal role in sustaining energy metabolism and redox balance during PVY infection.

Fig 2 illustrates that upon infection with the virus, a network of host proteins becomes activated in potato plants. Additionally, the virus proteins establish a network essential for successful infection. This observation aligns with the emerging 'network for network' theory in virus-host interactions, as proposed by a previous study [15].

The use of multiple algorithms to identify hub proteins is a standard approach in network analysis, as it minimizes potential biases from reliance on a single algorithm and increases the reliability of the results. In our study, we used four different algorithms (MNC, DMNC, DEGREE, and MCC) through Cyto-Hubba to identify hub proteins, which proved to be an effective strategy. Each of these algorithms uses a unique method to detect hub proteins, providing a broader range of results. MNC and DMNC focus on maximum cliques within the network, DEGREE considers the number of edges connected to a node, and MCC calculates the number of shortest paths through a node. By using algorithms with different approaches, we can capture multiple facets of the network and provide a more thorough understanding of hub proteins

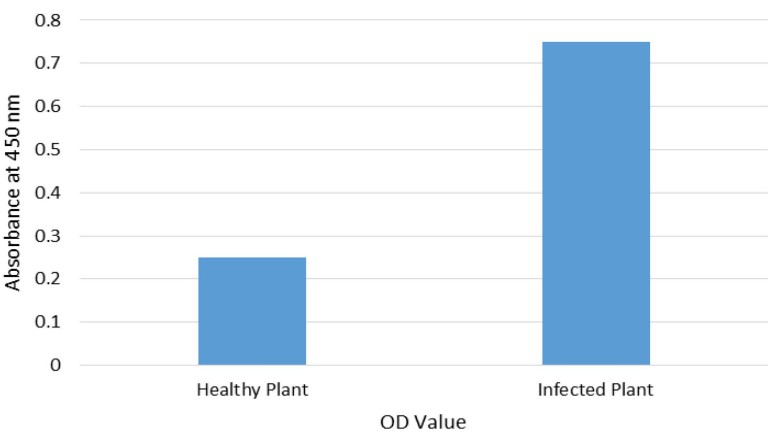

**Fig 1. Mean OD values of healthy and PVY-infected plants measured by ELISA (at 450 nm).** Data represents mean from three independent experiments; statistical differences determined by t-test (P < 0.05).

[16]. As described in the Materials and Methods section, 11 hub proteins were identified with the most interactions and significant roles in the network from all protein interactions (Fig 3). The detailed description of these hub proteins is provided in Table 2. All identified hub genes were upregulated during infection. However, certain genes may not show the highest expression levels, although they play a crucial role in the interaction between the virus and the host plant. PPI network analysis is a valuable tool for uncovering key genes involved in biological processes.

Integrated gene network analysis identified critical hub genes associated with the potato defense response against potato virus Y (PVY), primarily related to mitochondrial energy metabolism and redox regulation. Among these, ATP synthase subunits (e.g., Soltu.DM.07G008910.1, M1CAI4_SOLTU, and M1D096_SOLTU) emerged as central players, reflecting their essential role in maintaining cellular energy production during viral stress. Studies in *Nicotiana benthamiana* have shown that ATP synthase activity is upregulated during viral infections to meet the increased energy demands of antiviral responses [17].

NADH-ubiquinone oxidoreductase chain 3 (ND3), a central subunit of mitochondrial complex I, ranked high in the network, emphasizing its role in maintaining redox homeostasis. Complex I facilitates electron transfer in the mitochondrial electron transport chain and balances the production of reactive oxygen species (ROS) [18]. Excess ROS can damage cellular components, but controlled ROS bursts are critical for the activation of defense pathways, such as the expression of pathogenesis-related (PR) proteins. In Arabidopsis, disruption of the subunits of complex I led to increased susceptibility to viral pathogens due to impaired ROS signaling [19]. This is consistent with our results and suggests a dual role of ND3 in energy production and redox regulation during PVY infection.

In addition, NAD(P)-binding proteins of the Rossmann fold superfamily (e.g., Soltu.DM.07G015800.3, Soltu.DM.12G007550.1) were identified as key hubs. These proteins are involved in redox reactions, including the regeneration of antioxidants such as glutathione, which mitigate oxidative stress induced by viral infection. Rossmann fold domains have also been implicated in phytohormone signaling, particularly in defense pathways [20]. For example, in tomatoes, activation of SA biosynthesis by upregulation of PR1 and PAL genes can enhance resistance to viral infection and increase the expression of these genes [21]. Their high ranking in our network suggests a similar mechanism in potatoes, where these proteins may interface with SA signaling to limit PVY replication.

## Gene ontology and pathway enrichment analysis of subnetwork genes in potato plants infected with PVY

Genes within the subnetwork interact closely with hub genes, which is crucial in defining key biological pathways and cellular responses in PVY-infected potato plants. To gain deeper insights into these functional pathways, GO and KEGG

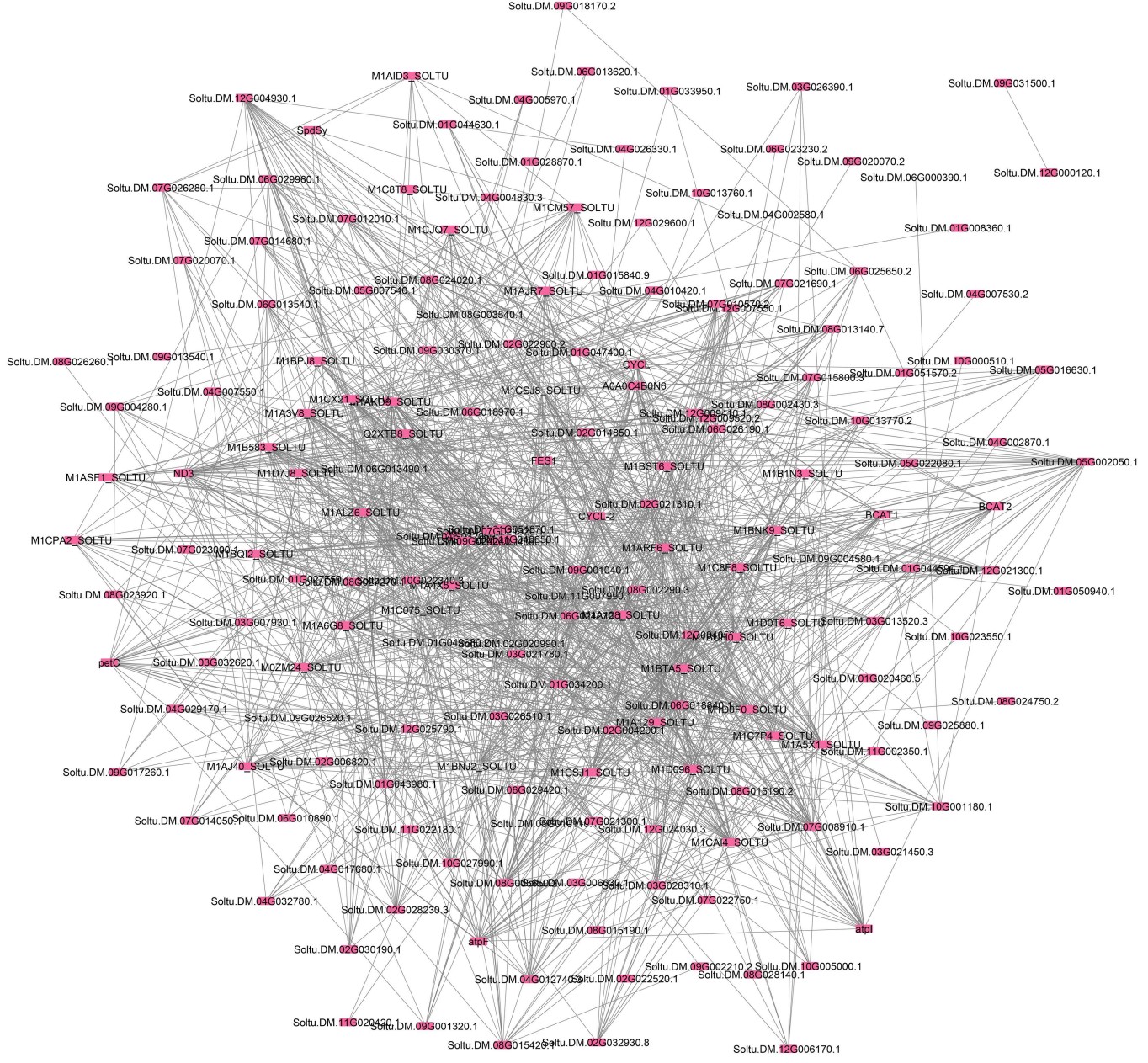

**Fig 2. PPI network of the up-and down-regulated genes in potato plants infected by PVY was constructed using Cytoscape software.**

pathway enrichment analyses were performed. GO provides a structured framework for classifying gene functions across three main categories: BP, MF, and CC. This study revealed substantial alterations in various biological categories and functions, including oxidative phosphorylation and glucosinolate biosynthetic processes, underscoring their central role in antiviral defense.

This study revealed substantial alterations in various biological categories and functions. Among the BP, notable up-regulation was observed in categories such as glucosinolate biosynthetic processes, cellular responses to stress,

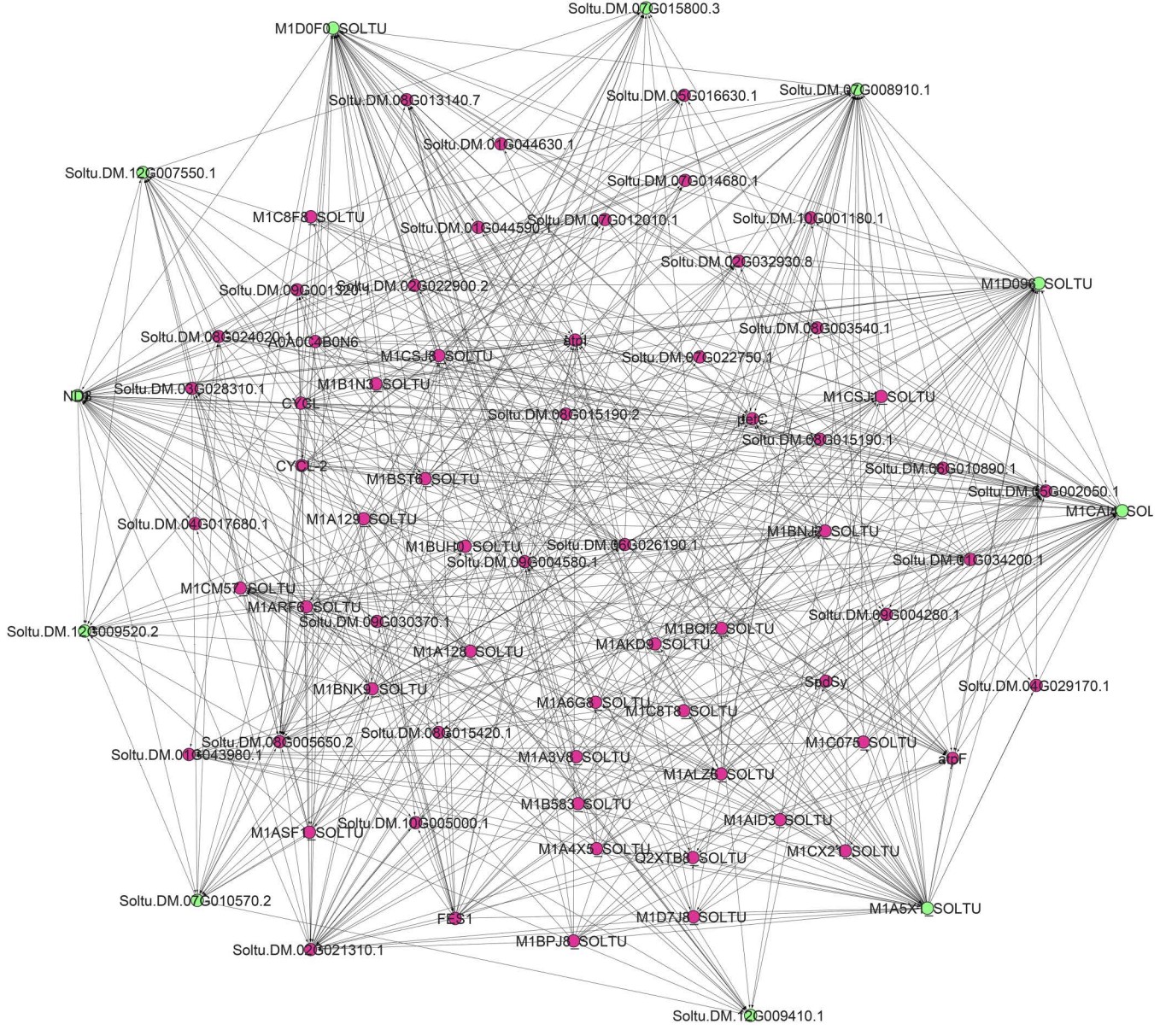

**Fig 3. Hub genes and subnetwork of the up-and down-regulated genes in potatoes infected by PVY, along with their known neighbors, based on data using the CytoHubba App.**

oxidative phosphorylation, carbohydrate derivative biosynthetic processes, fatty acid beta-oxidation, protein stabilization, and organonitrogen compound biosynthetic processes (Fig 4).

In terms of CC, significant enrichment was observed in categories such as the proton-transporting ATP synthase complex, the mitochondrial respiratory chain complex III, the chloroplast, and the cell periphery. Additionally, the cytoplasm exhibited the highest observed gene count, highlighting its critical role in various cellular processes (Fig 5).

Furthermore, an analysis of MF revealed that metal ion binding, translation initiation factor activity, ATP binding, iron-sulfur cluster binding, oxidoreductase activity, and proton transmembrane transporter activity were notably

**Table 2. Ranking of hub genes identified in PVY-infected potato using CytoHubba.**

| Rank | Gene ID | Ranking Method | Gene description |
|---|---|---|---|
| 1,3 | Soltu.DM.07G008910.1 | MCC, MNC, Degree | ATP synthase subunit |
| 2 | M1CAI4_SOLTU | MCC | ATP synthase subunit gamma, mitochondrial |
| 3,2 | M1D096_SOLTU | MCC, MNC, Degree | ATP synthase subunit beta, mitochondrial-like |
| 3,2 | M1A5X1_SOLTU | MCC, MNC, Degree | |
| 2 | M1D0F0_SOLTU | MCC, MNC, Degree | |
| 1 | ND3 | Degree | NADH-ubiquinone oxidoreductase chain 3; Core subunit of the mitochondrial membrane respiratory chain NADH dehydrogenase |
| 1 | Soltu.DM.07G015800.3 | DMNC | NAD(P)-binding Rossmann-fold superfamily protein |
| 3 | Soltu.DM.07G010570.2 | DMNC | |
| 3 | Soltu.DM.12G009520.2 | DMNC | |
| 2 | Soltu.DM.12G007550.1 | DMNC | |
| 3 | Soltu.DM.12G009410.1 | DMNC | |

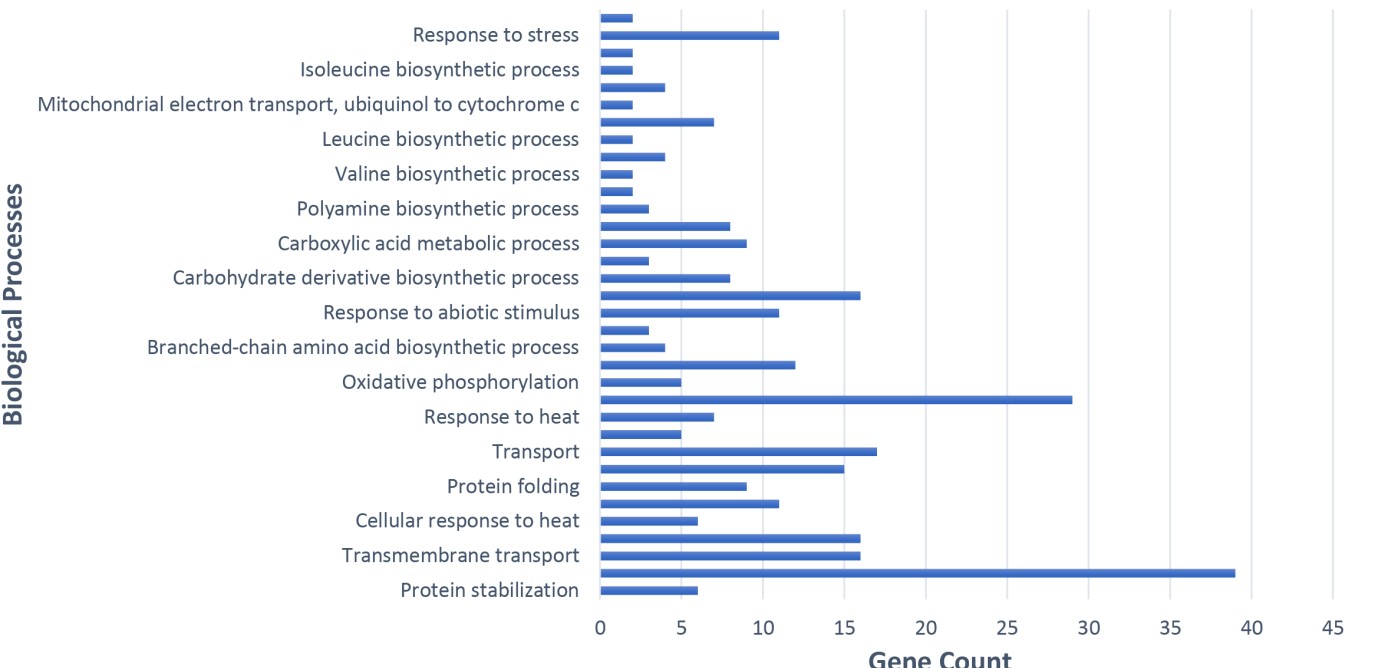

**Fig 4. Analysis of biological functions of key genes in potato infected with PVY, using Gene Ontology enrichment via STRING ver.10.**

up-regulated. Additionally, functions such as heterocyclic compound binding, electron transfer activity, unfolded protein binding, and ATP hydrolysis activity were found to be significant (Fig 6).

KEGG pathway analysis revealed significant enrichment in pathways such as metabolic pathways, oxidative phosphorylation, plant-pathogen interaction, biosynthesis of secondary metabolites, peroxisome, propanoate metabolism, fatty acid degradation, beta-alanine metabolism, protein processing in the endoplasmic reticulum, and many more. Metabolic pathways showed the highest frequency of observed genes, underscoring their critical role in the BP under study (Fig 7).

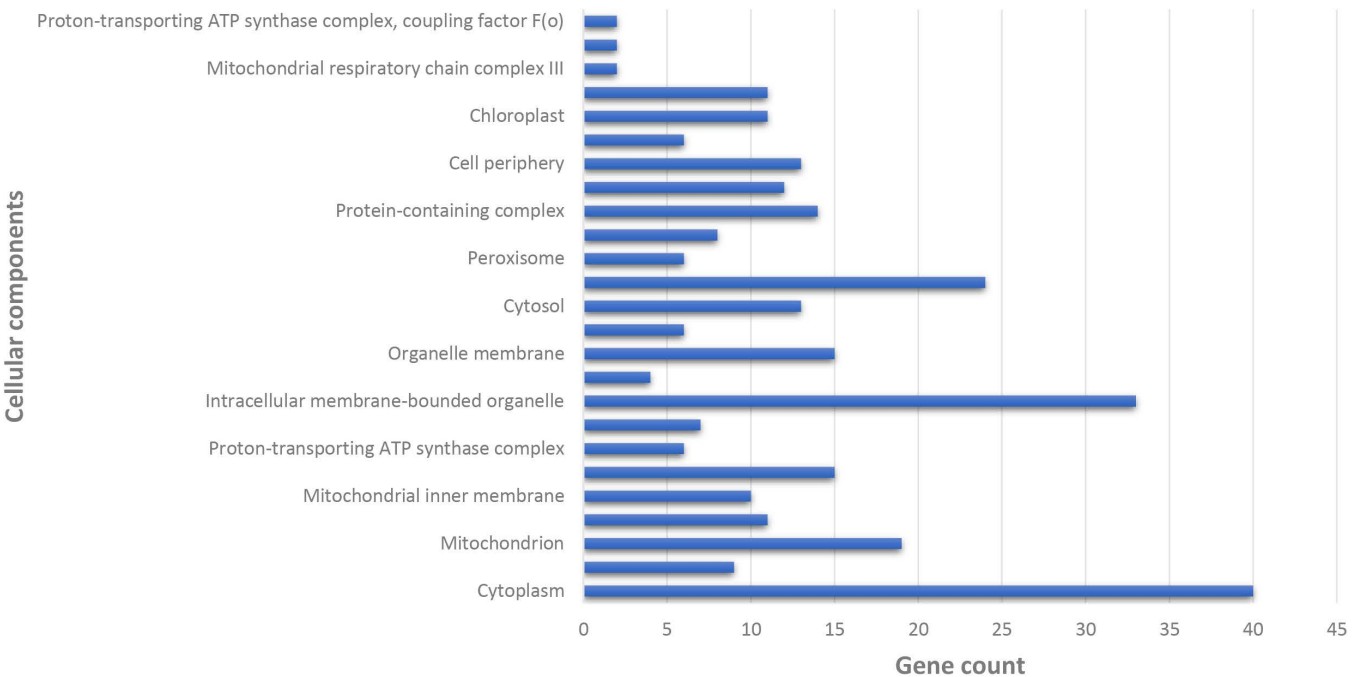

**Fig 5. Examination of cellular components of key proteins in potato infected with PVY, using Gene Ontology enrichment via STRING ver.10.**

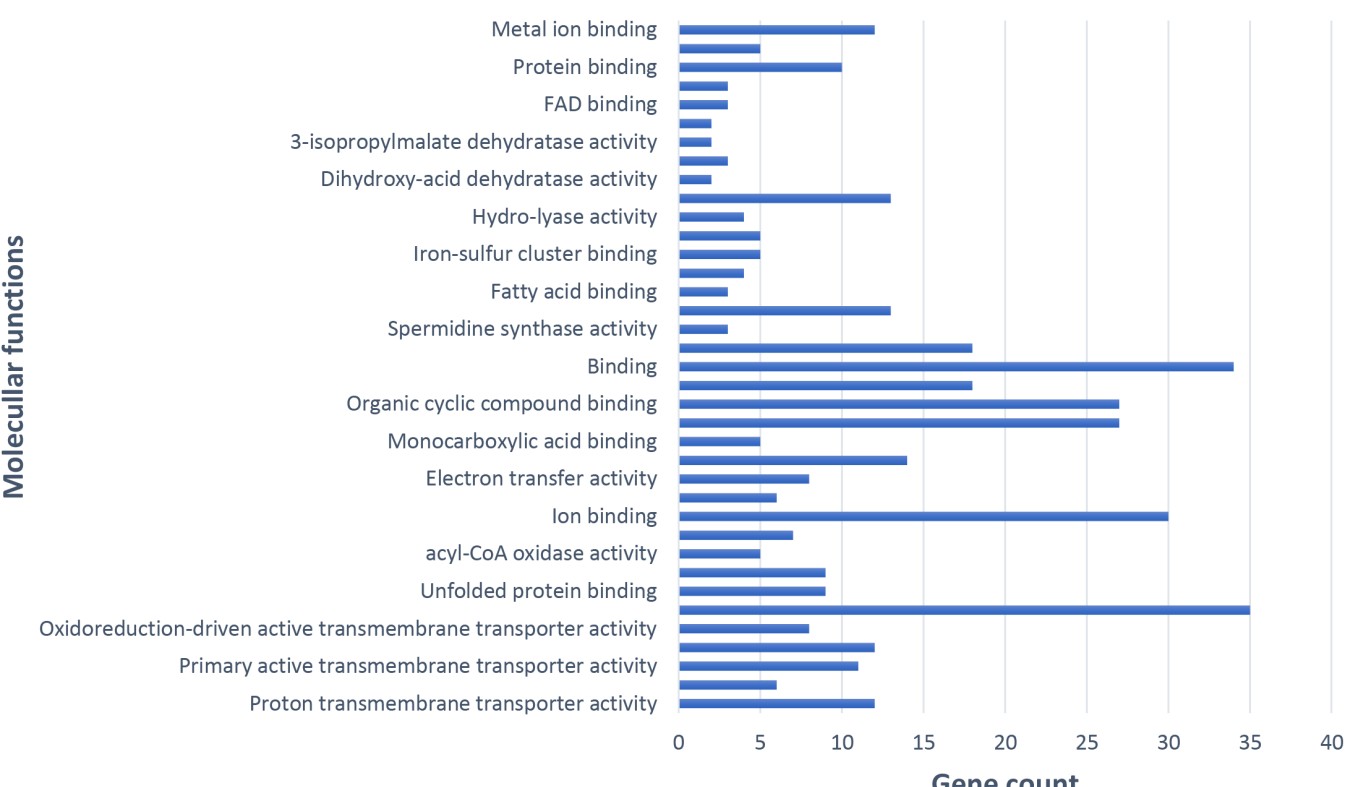

**Fig 6. Evaluation of molecular functions of key genes in potato infected with PVY, using Gene Ontology enrichment via STRING ver.10.**

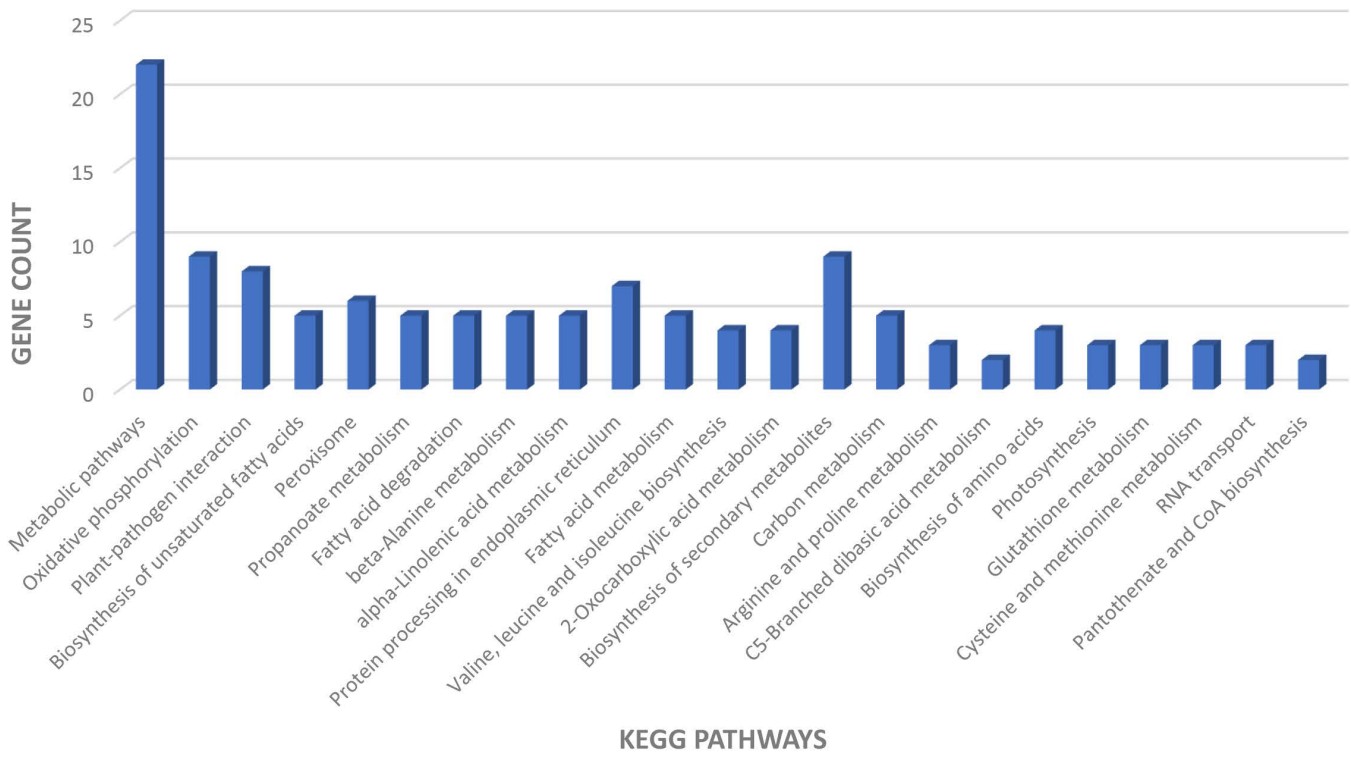

**Fig 7. Identification of biological pathways of key genes in potato response to PVY infection, based on KEGG analysis.**

The GO and KEGG pathway enrichment analyses provided valuable insights into the molecular responses of potato plants to PVY infection, highlighting significant changes in BP, CC, and MF. The strong interactions between hub genes and subnetwork genes highlight their regulatory roles in key pathways associated with the plant's adaptive response to viral stress. GO analysis revealed that stress-related pathways such as glucosinolate biosynthesis, oxidative phosphorylation, carbohydrate metabolism, and protein stabilization were significantly up-regulated. These results suggest that PVY infection triggers a complex molecular network involving the reprogramming of both primary and secondary metabolic processes to enhance plant defense mechanisms. For example, the increased expression of glucosinolate biosynthetic pathways has been linked to plant immunity, as these compounds act as key defense molecules against pathogens [22]. Furthermore, the activation of oxidative phosphorylation and carbohydrate metabolism, critical for energy production, suggests that PVY-infected cells allocate substantial energy reserves to mitigate viral stress. In addition, the upregulation of protein stabilization pathways implies a protective cellular response aimed at maintaining protein homeostasis and preventing protein misfolding or aggregation, which are common under stress conditions [23].

The analysis of CC and the enrichment analysis observed in key organelles such as chloroplasts and mitochondria highlight their critical role in energy metabolism and defense signaling during PVY infection. Mitochondrial respiratory pathways and chloroplast-associated functions play a pivotal role in regulating the levels of ROS, which act as key signaling molecules in plant immune responses [24]. The cytoplasm, which has the highest number of genes, underscores its importance in viral replication, protein translation, and defense-related molecular interactions. The MF analysis further corroborates these findings, revealing a notable upregulation in metal ion binding, ATP binding, and oxidoreductase activity, all of which are critical for maintaining cellular homeostasis and mitigating oxidative stress. The increased expression of translation initiation factors and ATP hydrolysis activity suggests that infected cells undergo adaptive changes to maintain

protein synthesis and energy availability under viral stress conditions [25]. KEGG pathway enrichment provided further insights, revealing the activation of pathways related to metabolic regulation, oxidative phosphorylation, plant-pathogen interactions, and secondary metabolite biosynthesis, all of which are integral to plant defense mechanisms. The strong enrichment of metabolic pathways indicates the need for cellular reprogramming to maintain energy production while redirecting resources toward antiviral defense. In addition, the activation of plant-pathogen interaction pathways suggests that PVY infection triggers distinct molecular defense mechanisms, including signal transduction pathways that regulate immune responses [26].

## Cluster analysis of the network

Cluster analysis of biological networks is a critical approach for identifying functional modules, predicting protein complexes, and detecting network biomarkers. It helps to uncover the underlying structure of biological networks. The choice of algorithms in CytoCluster depends on the specific needs of the user. In this research, six clustering algorithms were used. Among them, the IPCA algorithm, a density-based clustering method, was used to identify dense subgraphs within protein interaction networks. The algorithm calculates the weight of each edge by counting the common neighbors of the two connected nodes. The total weight of edges connected to a node determines its total weight, which in turn helps to identify the seed node. Initially, a seed node is treated as a cluster, and the IPCA algorithm recursively adds neighboring nodes to the cluster based on their priority. Two key conditions for adding a node to a cluster are the interaction probability of the node and the shortest path between the node and the existing cluster members [8].

This study employs cluster analysis of biological networks to identify key subnetworks and their associated functions. The top five clusters (ranks 1–5) were subjected to analysis based on the number of nodes, edges, and functional pathways (Table 3). The most interconnected cluster, designated as cluster rank 1, encompasses 23 nodes and 561 edges and is associated with critical functions such as oxidative phosphorylation, metabolic pathways, photosynthesis, and RNA transport. Clusters ranked 2–5 exhibited comparable numbers of nodes and edges; however, their specific functions are not detailed in the table, suggesting they may play complementary or supportive roles to cluster rank 1. The findings underscore the significance of cluster analysis in elucidating substantial biological pathways and interactions, with cluster rank 1 functioning as a pivotal nexus for critical cellular processes. The highest-ranked cluster (rank 1) exhibited essential functions such as oxidative phosphorylation, metabolic pathways, photosynthesis, and RNA transport. These functions align with pathways involved in other plant viral infections. For instance, oxidative phosphorylation and metabolic pathways are often disrupted during viral infections, as pathogens exploit the host's energy resources for replication [27]. A comparable association has been documented previously between RNA transport and viral propagation in potato-PVY interactions [28]. While clusters ranked 2–5 exhibited structural similarities to the top-ranked cluster, their specific functions remain to be elucidated, suggesting they may potentially support or regulate the primary pathways identified in cluster rank 1. These findings are consistent with research on PVY and other viral infections, where hub genes

**Table 3. Summary of the top-ranked clusters (1 to 5) identified by cluster analysis of expressed hub genes in PVY-infected potato plants using CytoCluster.**

| Cluster | Rank | Nodes | Edges | Function |
|---------|------|-------|-------|----------|
| 1 | 1 | 23 | 561 | Oxidative phosphorylation |
| 2 | 2 | 23 | 555 | Metabolic pathways |
| 3 | 3 | 22 | 537 | Photosynthesis |
| 4 | 4 | 22 | 543 | RNA transport |
| 5 | 5 | 20 | 483 | |

*All functions listed in this table are primarily proposed for cluster 1. Clusters 2–5 are considered to support the functions of cluster 1.

and interconnected networks are central to host-pathogen interactions [29]. This comparison underscores the conserved nature of specific pathways in viral infections.

## Promoter motif analysis of hub genes

The UFRs of hub genes spanning 1 kbp were analyzed to identify conserved motifs and consensus *cis*-regulatory elements (CREs). The UFR sequences were obtained from the Phytozome database. Using the MEME tool, six significant motifs ranging from 2 to 8 base pairs in length were identified in the promoter regions of the genes. GOMo analysis was performed to predict the biological roles of the transcription factor (TF) motifs, which were categorized into BP, MF, and CC. Predominant GO terms identified for BP included regulation of transcription, DNA-dependent processes, auxin-mediated signaling pathways, response to water deprivation, and responses to ethylene and abscisic acid stimuli. In terms of MF, these CREs were significantly enriched for transcription factor activity, kinase activity, and pseudouridine synthase activity. In addition, CC analysis highlighted their localization in cellular compartments such as the nucleus, plasma membrane, and endomembrane system (Table 4).

In this study, transcription factor activity emerged as a central molecular function across multiple motifs, underscoring the role of transcriptional reprogramming during viral infection. Plant viruses often hijack the host transcriptional machinery to suppress defense responses or redirect resources for viral replication. For example, motifs such as MA1267.1 and MA1268.1, which are enriched in nuclear- and plasma membrane-associated CC terms, may reflect viral strategies to manipulate nuclear-cytoplasmic trafficking or membrane-bound signaling nodes. Regulation of DNA-dependent

**Table 4. The conserved motifs were identified within the promoter regions of hub genes in PVY-infected potato plants among DEGs through MEME analysis.**

| Motif | logo | Predictions | Top 5 specific predictions |
|---|---|---|---|
| MA1161.1 | ATTTTAAAATTTAAA | 2 | MF transcription factor activity |
| MA1267.1 | AAAAAAAAAAAAAGTAAAAAAAAAAAAAAAA | 6 | MF transcription factor activity<br>CC nucleus<br>CC plasma membrane<br>BP regulation of transcription, DNA-dependent |
| MA1268.1 | AAAAAAAAAAAAAAAGAAAAAGTGAAAA | 7 | MF transcription factor activity<br>CC plasma membrane<br>CC nucleus<br>BP regulation of transcription, DNA-dependent<br>MF kinase activity |
| MA1281.1 | AAAAAAGTAAAAAAAAAAAAA | 8 | MF transcription factor activity<br>CC plasma membrane<br>BP regulation of transcription, DNA-dependent<br>CC nucleus<br>BP response to water deprivation |
| MA1379.1 | TTTTAAAATTTAAAT | 5 | MF transcription factor activity<br>CC endomembrane system<br>MF pseudouridine synthase activity<br>BP auxin-mediated signaling pathway |
| MA1380.1 | TTTTAAATTTTTTAA | 6 | MF transcription factor activity<br>BP response to ethylene stimulus<br>BP response to abscisic acid stimulus<br>BP response to salt stress<br>BP response to water deprivation |
| MA1385.1 | AAAAGAATATTTTAA | 4 | MF transcription factor activity<br>BP regulation of transcription<br>CC endomembrane system<br>CC plasma membrane |

transcription (MA1267.1, MA1268.1, MA1281.1) is consistent with documented viral tactics to alter host gene expression, including silencing of defense-related genes or promotion of viral mRNA synthesis [30].

Stress response pathways, including responses to water deprivation, salt stress, and phytohormones (ethylene and abscisic acid; MA1281.1, MA1380.1), have also been highlighted. Viral infections often exacerbate abiotic stress signals to destabilize host homeostasis and facilitate viral spread. For example, abscisic acid (ABA) and ethylene signaling are known to intersect with plant immunity, and their dysregulation can increase susceptibility to pathogens [31]. The enrichment of these terms suggests that the virus may use stress-responsive CREs to weaken host defenses or redirect metabolic resources. In particular, MA1379.1 was associated with auxin-mediated signaling and pseudouridine synthase activity. Auxin signaling is a common target for viral manipulation because it controls plant development and immune responses. Viruses often disrupt auxin pathways to promote symptom development or enhance viral movement [32]. Recent studies suggest that pseudouridylation can alter RNA-protein interactions, thereby affecting viral replication or host defense mechanisms. For example, functional characterization of pseudouridine synthase 4 during brome mosaic virus (BMV) infection in *N. benthamiana* revealed its role in binding to BMV-positive strand RNA, disrupting encapsidation, and ultimately leading to a reduction in viral RNA accumulation and systemic movement of BMV [33].

The CC terms, including the endomembrane system and plasma membrane (MA1379.1, MA1385.1), may reflect viral utilization of membrane networks for replication complex assembly or intracellular trafficking. Furthermore, kinase activity (MA1268.1) has been observed to implicate phosphorylation-dependent signaling cascades in viral pathogenesis, a mechanism that has been observed in plant-virus interactions where host kinases are co-opted to regulate viral replication [34].

## The miRNA target prediction for hub genes in potato infection response to PVY

One of the objectives of this study was to identify miRNAs that target hub genes that act as central regulators of potato infection mechanisms against PVY. For this purpose, the web-based tool psRNATarget was used to predict candidate miRNAs capable of interacting with these hub genes. Computational analysis revealed 28 potential miRNA-target pairs involving 16 unique miRNAs from evolutionarily conserved families (Fig 8). Notably, two hub genes, Soltu.DM.12G007550 and Soltu.DM.07G015800, were identified as the most frequently targeted, demonstrating regulation by multiple miRNAs. These genes, which are central to potato infection responses, exhibited complex regulatory interactions, suggesting their pivotal role in modulating PVY infection pathways.

The hub genes identified in this study, particularly NAD2, represent promising translational targets for enhancing PVY resistance in potato breeding programs. Their strong upregulation in resistant cultivars and centrality in mitochondrial redox signaling networks underscore their functional relevance. These genes could serve as molecular markers for marker-assisted selection (MAS) or be engineered using RNA interference (RNAi) or CRISPR/Cas9 gene editing to improve resistance traits. Previous studies have demonstrated the successful use of RNAi in suppressing susceptibility genes and enhancing viral resistance in Solanaceae crops [33,35]. Moreover, miRNAs regulating these hub genes offer another layer of intervention, as shown by the modulation of AP2/ERF targets by miR172 in tomato late blight resistance [36]. The identification of codon usage patterns optimized for expression provides a foundation for future strategies such as codon deoptimization, which has been proposed to reduce viral fitness [37]. These insights collectively establish a pipeline for translating gene network discoveries into breeding and biotechnological solutions for PVY management.

The role of miRNAs in regulating plant responses to biotic and abiotic stresses is a rapidly evolving area of research. For instance, miRNAs such as stu-miR5303, stu-miR8019, and stu-miR171 identified in the StKNOX study regulate developmental pathways, nutrient homeostasis, and stress responses, including late blight infection (caused by *Phytophthora infestans*), heat, cold, drought, and hypoxia [38]. Also, research has shown that overexpression of miR172a and miR172b in tomatoes enhances resistance to late blight by suppressing AP2/ERF transcription factor genes [36]. This finding suggests that miR172 may have a similar function in potato defense mechanisms against PVY, possibly by modulating AP2/ERF transcription factors to enhance disease resistance. Moreover, the complex regulatory networks surrounding these hub genes underscore their potential as primary targets for genetic engineering or breeding initiatives aimed at enhancing

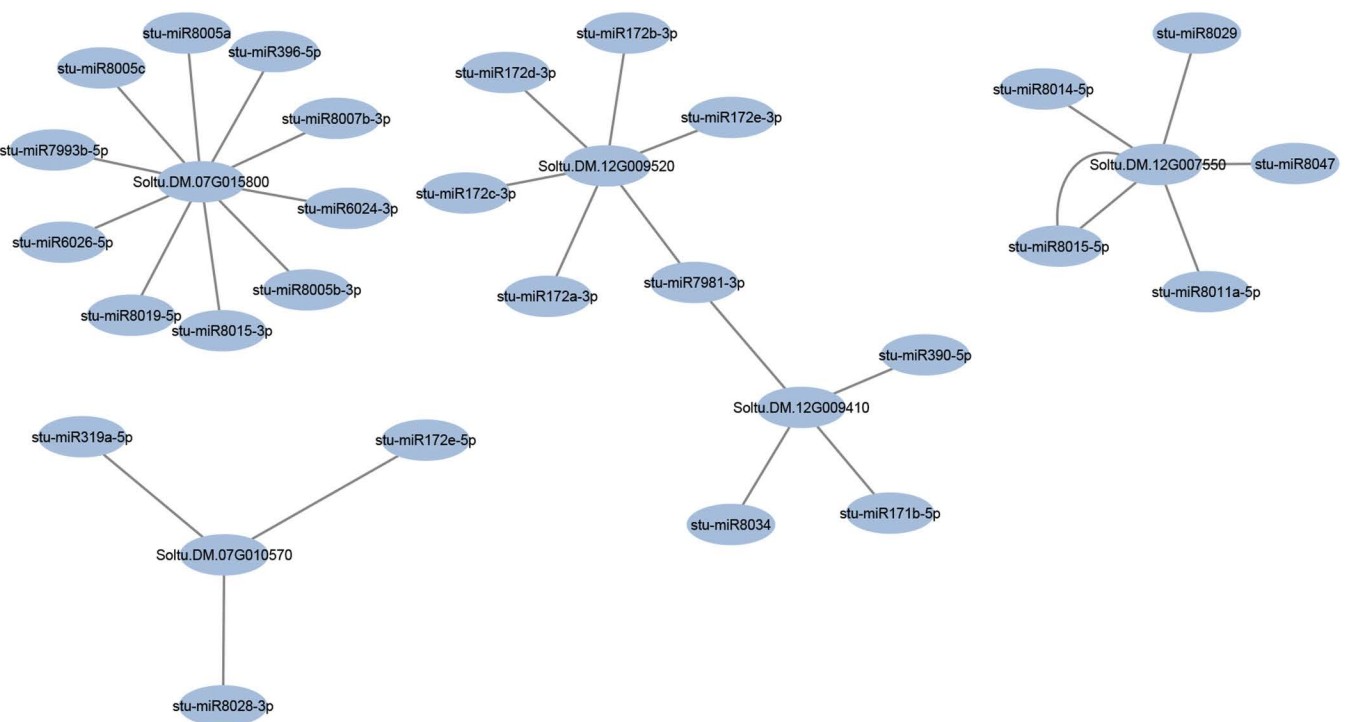

**Fig 8. The miRNA targeting analysis for PVY-infected potato hub genes.**

PVY resistance. The modulation of the expression of these genes or their corresponding microRNAs could serve as a pathway to reinforce the defense mechanisms of potatoes. One potential avenue for achieving this objective is the employment of CRISPR/Cas9 or RNA interference (RNAi) technologies to suppress negative regulators of defense pathways [39]. So, these genes emerge as highly valuable targets for further functional investigations or genetic engineering strategies aimed at enhancing resistance to PVY. Our discovery of multi-miRNA regulation of potato hub genes aligns with findings in tomatoes, where miR172 targets AP2/ERF factors to enhance late blight resistance [35]. Similarly, conserved miRNAs like stu-miR5303, linked to stress responses in potatoes [40], highlight evolutionary conservation in Solanaceae immunity. Unlike single miRNA-target models, our network-based approach underscores combinatorial miRNA control.

## Codon usage analysis of hub genes targeted in potatoes against PVY

Codon usage bias analysis of hub genes in potatoes against PVY provided detailed insights into their translational efficiency and evolutionary adaptation. Metrics, GC content, ENC, CAI, and RSCU, were used to analyze the patterns of codon preference. The GC content of the analyzed genes ranged from 35% to 40%, highlighting a moderate GC bias across the dataset. ENC values varied from 51–52, reflecting a moderate codon usage bias influenced by both mutational pressure and natural selection. The CAI values for these genes peaked at around 0.7, indicating a high degree of adaptation to the potato translational machinery (Fig 9). Genes with lower ENC values showed stronger codon preferences, highlighting their functional importance in potato defense mechanisms. Genes with higher CAI values are likely optimized for increased expression of critical proteins required for an effective defense response [41].

RSCU analysis revealed distinct patterns of codon usage among critical amino acids. For phenylalanine (Phe), the TTT codon showed strong preference (69.84% frequency, 87 counts) and optimal adaptiveness (w_cai = 1), in contrast to the underrepresented TTC (30.16%, w_cai = 0.41). Leucine (Leu) favored TTG (36.51%, w_cai = 0.83), while TTA was

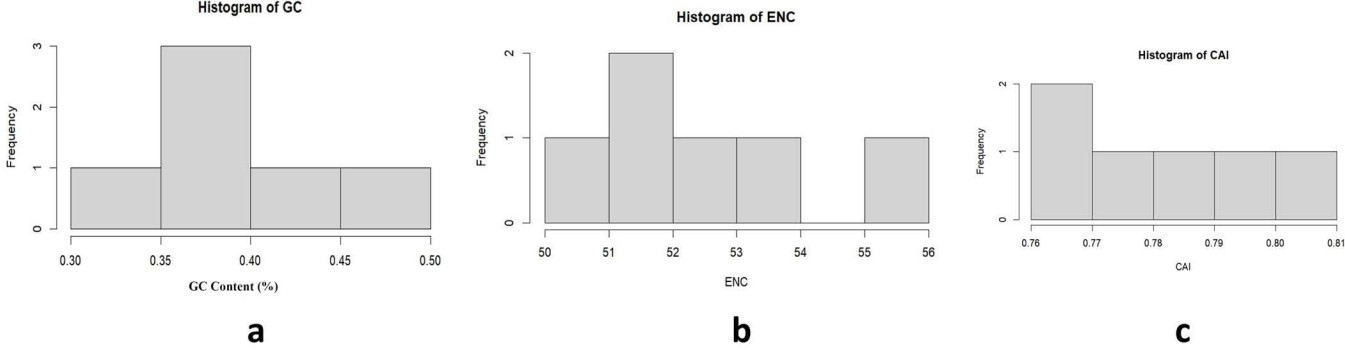

**Fig 9. Codon usage bias analysis for potato hub genes against PVY. (a)** GC content, **(b)** Effective Number of Codons (ENC), and **(c)** Codon Adaptation Index (CAI), highlighting key metrics of codon preference and their implications for translational efficiency.

rare (2.46%). Serine (Ser) showed equal dominance for TCT and TCA (34.95% each, w_cai = 1), with TCG being the least efficient (12.26%, w_cai = 0.30). Tyrosine (Tyr) showed high efficiency for TAT (51.83%, w_cai = 1) versus underutilized TAC (24.14%, w_cai = 0.31), and cysteine (Cys) was overwhelmingly biased toward TGT (100%, w_cai = 1) versus TGC (28.75%, w_cai = 0.40). The w_cai analysis highlighted codons with w_cai = 1 (e.g., TTT, TTG, TCT/TCA, TAT, TGT) as highly adaptive, aligning with host tRNA pools for efficient translation, whereas codons with w_cai < 0.5 (TTC, TAC, TGC) suggested constraints due to mutational pressure or host adaptation (Table 5). The critical role of codons in shaping plant evolution has been emphasized in previous studies of codon usage in potatoes [37]. In practice, these results could guide antiviral strategies such as synonymous codon deoptimization, in which preferred codons are replaced by rare ones to reduce viral fitness. This approach offers a potential route to the development of potato cultivars resistant to PVY. The analysis highlights the complex interplay between mutation, host adaptation, and translational efficiency in shaping viral codon usage and lays the groundwork for genetic engineering and breeding initiatives aimed at enhancing viral resistance.

**Table 5. Analysis of Relative Synonymous Codon Usage (RSCU) and Codon Adaptation Index (w_cai) for potato hub genes involved in PVY.**

| aa_code | Amino Acid | Codon | Subfamily | Counts (cts) | Proportion | w_cai |
|---|---|---|---|---|---|---|
| 1 | Phe (F) | TTT | Phe_TT | 87 | 0.6984 | 1 |
| 2 | Phe (F) | TTC | Phe_TT | 37 | 0.3016 | 0.4138 |
| 3 | Leu (L) | TTA | Leu_TT | 3 | 0.0246 | 0.0833 |
| 4 | Leu (L) | TTG | Leu_TT | 46 | 0.3651 | 0.8333 |
| 5 | Ser (S) | TCT | Ser_TC | 54 | 0.3495 | 1 |
| 6 | Ser (S) | TCC | Ser_TC | 20 | 0.1295 | 0.3273 |
| 7 | Ser (S) | TCA | Ser_TC | 54 | 0.3495 | 1 |
| 8 | Ser (S) | TCG | Ser_TC | 19 | 0.1226 | 0.3064 |
| 9 | Tyr (Y) | TAT | Tyr_TA | 65 | 0.5183 | 1 |
| 10 | Tyr (Y) | TAC | Tyr_TA | 20 | 0.2414 | 0.3182 |
| 11 | Cys (C) | TGT | Cys_TG | 45 | 1 | 1 |
| 12 | Cys (C) | TGC | Cys_TG | 20 | 0.2875 | 0.4059 |
| 13 | Trp (W) | TGG | Trp_TG | 45 | 1 | 1 |

## Analysis of real-time PCR results

The expression levels of three hub genes (*NAD1*, *NAD2*, and *NAD3*) were analyzed in PVY-infected potato cultivars, Sante (resistant) and Agria (susceptible), using real-time PCR. The most significant finding was the 5.58-fold upregulation of *NAD2* in the resistant Sante cultivar. Moreover, *NAD1* exhibited consistent expression, with fold changes of 1.82. Conversely, *NAD3* demonstrated expression with a fold change of 3.01, indicating its involvement in specific regulatory pathways that may be activated during the infection stage or in response to tissue-specific signals. In the susceptible cultivar Agria, *NAD1* and *NAD2* exhibited expression, with fold changes of 0.65 for *NAD1* and 0.62 for *NAD2*, suggesting a weakened defense response that could contribute to susceptibility. *NAD3* exhibited expression with a fold change of 1.04, further underscoring the differential regulation of hub genes between resistant and susceptible cultivars (Fig 10).

The observed differences in gene expression between the resistant Sante and susceptible Agria cultivars provide important clues to the molecular basis of plant resistance to PVY. The consistent up-regulation of *NAD1* and *NAD2* in Sante suggests that these genes play a key role in activating defense mechanisms such as oxidative phosphorylation and stress signaling. These processes are critical for the generation of ROS and other signaling molecules that help fight the invading pathogen [42]. The increased expression of NAD2 in PVY-resistant Sante cultivars mirrors the mitochondrial ROS amplification observed in Arabidopsis rtp7 mutants during immune responses. Similar to the pathogen effector Avr-Pita in rice, which suppresses mitochondrial ROS by targeting OsCOX11, PVY may employ analogous virulence strategies to disrupt ROS-mediated defenses in susceptible cultivars. The significant increase in NAD2 expression in Sante is consistent with findings in *Sclerotinia sclerotiorum* infections, where mitochondrial ROS accumulation via QCR8 dysfunction promotes resistance [43]. This suggests that redox hubs play a conserved role across pathosystems. Unlike studies focusing on effector-mediated ROS suppression, however, our work highlights host-driven ROS potentiation as a critical resistance mechanism, advancing the paradigm of mitochondrial redox dynamics in plant-virus interactions.

The higher expression levels of *NAD3* in Sante further support the notion that multiple genes contribute to a coordinated defense response, ensuring a robust and effective immune response against PVY. On the other hand, the downregulation of *NAD1* and *NAD2* in Agria indicates a failure to mount an adequate defense response, resulting in increased susceptibility to PVY infection. The minimal upregulation of *NAD3* in this cultivar underscores the importance of gene expression patterns in determining plant resistance. These findings have significant implications for agricultural practice, as they suggest that manipulating the expression of key defense genes could enhance plant resistance to viral pathogens. Further research is needed to elucidate the precise mechanisms by which these genes regulate plant immunity and to develop targeted strategies to improve plant resistance [44]. By understanding and exploiting these genetic factors, we may be able to develop more resilient potato varieties that are better equipped to withstand PVY infection and other viral threats. While this study

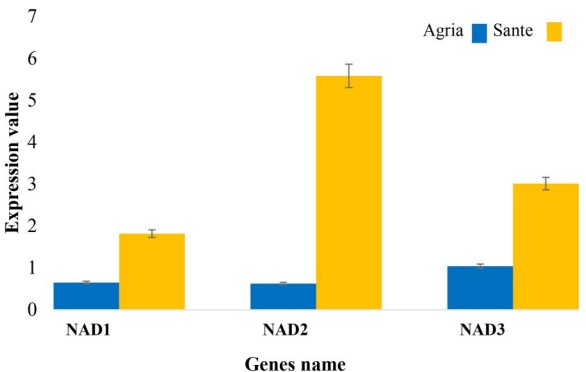

**Fig 10. Differential expression (p-value 0.05) of *NAD1*, *NAD2*, and *NAD3* in resistant (Sante) and susceptible (Agria) potato cultivars post-PVY infection.**

identified novel hub genes and regulatory networks underlying PVY resistance in Sante, it is critical to contextualize these findings with known resistance genes like Rysto. The Rysto gene, encoding a TIR-NLR immune receptor, confers extreme resistance to PVY in certain cultivars. Although earlier studies postulated Rysto's presence in Sante [7], our current work focused on uncharacterized pathways and did not experimentally validate Rysto in this cultivar. Future studies should combine targeted sequencing of Rysto with functional assays to clarify its contribution to Sante's resistance. Comparative analysis of Rysto-positive and Rysto-negative cultivars could disentangle shared and unique resistance mechanisms, such as the NAD2-mediated redox regulation identified here. Also, another study reported gene expression changes during the first 12 hours after PVY infection in Sante [45]. Although their data were not directly analyzed in our current work, our findings provide complementary insights into resistance mechanisms by focusing on persistent hub genes (e.g., NAD2) and integrating regulatory layers (e.g., miRNAs and codon usage). Future integrated analysis of early and late-phase transcriptome data could elucidate the synergy between immediate and sustained defense responses.

This study advances our understanding of the molecular pathways underlying PVY resistance in potatoes. However, several constraints warrant consideration. First, relying on existing RNA-seq data from various sources introduces potential confounding factors, such as discrepancies in experimental design (e.g., pathogen isolates and growth conditions) and genetic backgrounds of cultivars. While qPCR validation of pivotal genes such as NAD2 strengthened our conclusions, subsequent research should prioritize de novo transcriptomic profiling under standardized infection protocols to resolve cultivar-specific responses. Second, although greenhouse assays are valuable for providing controlled mechanistic insights, they fail to account for ecological variables prevalent in agricultural fields, such as aphid transmission efficiency, soil microbiota, and abiotic stress. To address this, we recommend multi-season field evaluations of genetically engineered lines (e.g., CRISPR-modified NAD2 variants or miRNA-overexpressing cultivars) in PVY-endemic zones to evaluate the durability of resistance and the agronomic performance. Despite these limitations, our multi-omics strategy identifies evolutionarily conserved nodes (e.g., NAD2 redox signaling and AP2/ERF transcriptional cascades) with translational potential for enhancing specific traits in potato breeding.

## Conclusion

This study integrated computational network analysis and experimental validation to identify key hub genes governing potato defense against PVY. These genes, central to mitochondrial energy metabolism and redox regulation, were significantly upregulated in the resistant cultivar Sante (*NAD1*: 1.82-fold, *NAD2*: 5.58-fold, *NAD3*: 3.01-fold) but downregulated in the susceptible Agria, highlighting their role in antiviral responses. Enrichment analysis of pathways revealed oxidative phosphorylation, glucosinolate biosynthesis, and ROS signaling as critical defense mechanisms. A subsequent analysis of promoter motifs identified stress- and phytohormone-responsive elements as being linked to transcriptional reprogramming. Moreover, the targeting of microRNAs and codon usage bias were identified as key elements of post-transcriptional regulation. The findings of this study provide actionable targets for the development of PVY-resistant cultivars via CRISPR/Cas9 or RNAi strategies, emphasizing metabolic adaptation and redox homeostasis as pillars of plant immunity.

## Supporting information

**S1 File. Differentially expressed genes shared between PVY-infected Payette Russet plants as compared to mock-infected plants at 24 hpi and Russet Burbank at 1 wpi and 4 wpi as compared to mock-infected plants (Benjamini and Hochberg adj. p-value < 0.05).**
(XLSX)

## Author contributions

**Conceptualization:** Abozar Ghorbani.

**Data curation:** Roya Karimipour.

**Methodology:** Roya Karimipour, Masoud Naderpour.

**Software:** Abozar Ghorbani.

**Supervision:** Abozar Ghorbani, Davoud Koolivand.

**Validation:** Roya Karimipour, Mahsa Rostami.

**Writing – original draft:** Roya Karimipour.

**Writing – review & editing:** Abozar Ghorbani, Davoud Koolivand, Masoud Naderpour, Mahsa Rostami.

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
