## [Decision Letter · Decision Letter 0]

15 May 2025

Dear Dr. Ghorbani,

Thank you for submitting your manuscript to PLOS ONE. After careful consideration, we feel that it has merit but does not fully meet PLOS ONE’s publication criteria as it currently stands. Therefore, we invite you to submit a revised version of the manuscript that addresses the points raised during the review process.

We look forward to receiving your revised manuscript.

Kind regards,

Rajarshi Gaur

Academic Editor

PLOS ONE

Additional Editor Comments (if provided):

Reviewers' comments:

Reviewer's Responses to Questions

**Comments to the Author**

1. Is the manuscript technically sound, and do the data support the conclusions?

Reviewer #1: Yes

Reviewer #2: Partly

2. Has the statistical analysis been performed appropriately and rigorously?

Reviewer #1: Yes

Reviewer #2: No

3. Have the authors made all data underlying the findings in their manuscript fully available?

Reviewer #1: Yes

Reviewer #2: Yes

4. Is the manuscript presented in an intelligible fashion and written in standard English?

Reviewer #1: Yes

Reviewer #2: Yes

Reviewer #1: Critical Review and Recommendations

1. Research Design

Strengths: The study effectively integrates bioinformatics (PPI networks, GO/KEGG analysis) with experimental validation (qPCR) to identify hub genes in potato-PVY interactions. Use of multiple algorithms (MCC, Degree, DMNC, MNC) to identify hub genes enhances robustness.

Recommendations (if possible): Include time-course data to track dynamic gene expression changes post-infection. Compare more resistant/susceptible cultivars to generalize findings.

Writing (Methods, Results, Discussion, Grammar)

Strengths: Clear structure and comprehensive coverage of analyses. Results are well-supported by figures/tables.

Recommendations:

Methods: Specify RNA-seq parameters (read depth, alignment tools) for reproducibility. Clarify statistical tests used for ELISA/qPCR (e.g., ANOVA for multi-group comparisons).

Results: Avoid redundancy between text and figures (e.g., Table 2 and Figure 3 overlap). Highlight key findings upfront (e.g., NAD2’s 5.58-fold change in Sante).

Discussion: Contrast results with prior studies (e.g., link NAD2 to ROS signaling in other crops). Address limitations (e.g., reliance on public RNA-seq data, lack of field validation).

Grammar/Clarity: Fix minor errors (e.g., "lacuna persists" → "gap remains"; "mitochondrial linear membrane" → "mitochondrial inner membrane"). Simplify complex sentences (e.g., break down the 60-word abstract sentence).

Data Presentation

Strengths: Effective use of networks (Figures 2–3) and enrichment plots (Figures 4–7). Codon usage metrics (Figure 9) are novel and well-visualized.

Recommendations:

Alternative Visualizations:

a) Heatmaps: Show expression trends of hub genes across cultivars/timepoints.

b) Volcano Plots: Highlight DEGs from RNA-seq data (if available).

c) Pathway Maps: Annotate KEGG pathways (e.g., oxidative phosphorylation) with gene expression overlays.

Figure Improvements:

• Label axes clearly in Figure 9 (e.g., "GC Content (%)").

• Include p-values or confidence intervals in Figure 10 (error bars are unclear).

Tables:

• Merge Tables 2 and 3 (hub genes and clusters) to reduce redundancy.

• Add a summary table for miRNA-gene interactions (Figure 8).

Overall Suggestions

a. Reproducibility: Share code/scripts for bioinformatics analyses (e.g., Cytoscape workflows).

b. Impact: Discuss translational applications (e.g., breeding targets, RNAi strategies) earlier in the discussion.

c. Supplementary Data: Provide raw qPCR Ct values and primer validation data.

Final Note: The study is rigorous and impactful but would benefit from deeper mechanistic insights and clearer data storytelling. Major revisions would elevate its clarity and translational relevance.

Reviewer #2: I identified three major issues that require significant changes and improvement before the article can be accepted for publication. The article is analysing previously obtained and published data of Ross et al. (Viruses 2022, 14(3), 523; https://doi.org/10.3390/v14030523) who sequenced the transcriptomes of PVY-resistant cv. Payette Russet and susceptible cv. Russet Burbank. After bioinformatics analyses, the authors identified in this dataset “hub genes” essential for potato defence against PVY and then validated their differential expression in resistant cv. Sante and susceptible cv. Agria. PVY resistance described in literature so far is a simple trait encoded by single resistance genes such as Rysto, Ryadg and Rychc. 1. The Authors did not mention which of these genes is present in their material (Sante) and if it is also underlying the resistance of Payette Russet. This is an essential information missing. No that the Rysto gene is sequenced (Grech-Baran M, Witek K, Szajko K, Witek AI, Morgiewicz K, Wasilewicz-Flis I, Jakuczun H, Marczewski W, Jones JDG, Hennig J. Extreme resistance to Potato virus Y in potato carrying the Rysto gene is mediated by a TIR-NLR immune receptor. Plant Biotechnol J. 2020;18:655–67.), the Authors should validate experimentally if it is present in cv. Sante as postulated in some earlier works (Flis et al. 2005, Baebler at al. 2009).

2. Another omission is not mentioning the work:

Baebler S, Krecic-Stres H, Rotter A, Kogovsek P, Cankar K, Kok EJ, Gruden K, Kovac M, Zel J, Pompe-Novak M, Ravnikar M. PVY(NTN) elicits a diverse gene expression response in different potato genotypes in the first 12 h after inoculation. Mol Plant Pathol. 2009 Mar;10(2):263-75. doi: 10.1111/j.1364-3703.2008.00530.x

In this work a transcriptome of cv Sante (inoculated or not with PVY) is described and this data set would be more relevant for the goal of this manuscript. Possibly, if the resistance in both Sante and Payette Russet is encoded by the same resistance gene, both transcriptome data sets could be analysed.

3. The experimental part of the article, i.e. qPCR validation of the differential expression of the hub genes is poorly described and analysed.

For the guidelines on how to perform and describe qPCR experiments, please refer to:

Bustin SA, Benes V, Garson JA, Hellemans J, Huggett J, Kubista M, Mueller R, Nolan T, Pfaffl MW, Shipley GL, Vandesompele J, Wittwer CT. The MIQE guidelines: minimum information for publication of quantitative real-time PCR experiments. Clin Chem. 2009 Apr;55(4):611-22. doi: 10.1373/clinchem.2008.112797

The results of the qPCR experiments should be analysed statistically to indicate the level of variation between biological and technical replicates and to demonstrate statistical significance of the observed differences in the genes’ expression.

Figure 10: legend is missing (which cultivar is shown in blue and which in yellow?), units on the y-axis are missing, error bars and indication of the statistically significant differences are missing.

Some minor corrections:

Line 27: a critical global crop – unclear; why is it critical? Please rephrase

Line 27: Potato Virus Y (PVY) - incorrect spelling, “virus” should not be capitalized. The correct spelling is: Potato virus Y

Line 31: and identified hub genes central to defense responses were constructed – unclear: while networks may have been “constructed”, the hub genes were rather identified than “constructed”

Lines 50-53 Potato Tuber Necrotic Ringspot Disease (PTNRD) caused by virus PVYNTN or PVYN should be also mentioned here

**Do you want your identity to be public for this peer review?** For information about this choice, including consent withdrawal, please see our Privacy Policy

Reviewer #1: No

Reviewer #2: No

---

## [Author Response · Author response to Decision Letter 1]

26 May 2025

April 1, 2025

Prof. Rajarshi Gaur

Academic Editor

PLOS ONE

Dear Prof. Rajarshi Gaur,

First, my co-authors and I would like to thank you and the two anonymous reviewers for considering our manuscript PONE-D-25-15213 with a positive attitude. We are very grateful to the reviewers for their comments, which clearly helped us to improve the quality of this manuscript. We agree with the reviewers on many points and have listed our response point by point at the end of this letter to explain why.

We hope that both you and the reviewers appreciate our revised text, and we are always available if you need further clarification on the revised text. For your information, all changes in the text are marked as "Track changes.

We thank you again for your time and attention and look forward to receiving the final decision on the manuscript.

With my best regards

Reviewers' Comments to Author:

Reviewer: 1

1. Research Design

Strengths: The study effectively integrates bioinformatics (PPI networks, GO/KEGG analysis) with experimental validation (qPCR) to identify hub genes in potato-PVY interactions. Use of multiple algorithms (MCC, Degree, DMNC, MNC) to identify hub genes enhances robustness.

Recommendations (if possible): Include time-course data to track dynamic gene expression changes post-infection. Compare more resistant/susceptible cultivars to generalize findings.

Response: In the current study, we focused on analyzing gene expression at the onset of visible symptoms (one week post-inoculation) to capture the host’s early transcriptional response to PVY infection. While this approach allowed us to identify key defense-related hub genes activated during symptom development, we fully agree that tracking expression dynamics at shorter intervals (e.g., 24, 48, 72 hours post-infection) would provide deeper insights into transient defense mechanisms. This is now a high priority for our future studies, and we are actively planning to incorporate time-course profiling into upcoming research. These investigations will be an integral part of our follow-up work to unravel the temporal regulation of defense pathways in potato-PVY interactions.

The cultivars Sante (resistant) and Agria (susceptible) were selected based on their well-documented contrasting responses to PVY in field conditions (Reference 5). In line with your suggestion, expanding this analysis to additional resistant and susceptible cultivars will be a priority in our future research plans to validate the universality of the identified hub genes and regulatory mechanisms across diverse genetic backgrounds.

The reference "5" was added to the manuscript.

2- Writing (Methods, Results, Discussion, Grammar)

Strengths: Clear structure and comprehensive coverage of analyses. Results are well-supported by figures/tables.

Recommendations:

Methods: Specify RNA-seq parameters (read depth, alignment tools) for reproducibility. Clarify statistical tests used for ELISA/qPCR (e.g., ANOVA for multi-group comparisons).

Response: The transcriptomic data analyzed in this study were obtained from a previously published dataset (Reference 7)

-The statistical details for the ELISA analysis were initially included in "Figure 1 caption" (as noted in our original submission), we fully agree that explicitly stating these parameters in the "Materials and Methods section" enhances reproducibility. In response to your comment, we have now added the following statement to the ELISA subsection of the Methods: "Statistical significance between PVY-infected and control groups was determined using Student’s t-test (P < 0.05). Three biological replicates were analyzed per group."

-As requested, we have updated the Materials and Methods section under Virus inoculation and hub gene expression analysis in potato cultivars to explicitly state: "A two-way ANOVA was performed to assess differences in gene expression between groups. Following a significant interaction effect (P* < 0.05), Tukey’s HSD post-hoc test was applied for pairwise comparisons to identify specific differences between resistant (Sante) and susceptible (Agria) cultivars under PVY-infected and control conditions."

3- Results: Avoid redundancy between text and figures (e.g., Table 2 and Figure 3 overlap). Highlight key findings upfront (e.g., NAD2’s 5.58-fold change in Sante).

Discussion: Contrast results with prior studies (e.g., link NAD2 to ROS signaling in other crops). Address limitations (e.g., reliance on public RNA-seq data, lack of field validation).

Response: Thank you for your insightful feedback. To address redundancy between the text and figures, we clarified that Figure 3 visually summarizes the co-expression network and hub gene positions, while Table 2 provides full quantitative details for reproducibility. Removing Table 2 would overwhelm the text with numerical data, so we retained it for clarity. Also, we confirm that all figures and tables were rigorously cross-checked to avoid textual overlap.

-As requested, we added these sentences to the "Analysis of Real-Time PCR Results" section: "The most significant finding was the 5.58-fold upregulation of NAD2 in the resistant Sante cultivar."

- These sentences were added to the "Gene Ontology and pathway enrichment analysis of subnetwork genes in potato plants infected with PVY" section as highlighting our finding:

This study revealed substantial alterations in various biological categories and functions, including oxidative phosphorylation and glucosinolate biosynthetic processes, underscoring their central role in antiviral defense.

- Also, we add these sentences to the "Analysis of PPI Networks and Identification of Hub Genes" section. : Integrated network analysis identified ATP synthase subunits and mitochondrial complex I components as top-ranked hub genes, underscoring their pivotal role in sustaining energy metabolism and redox balance during PVY infection

-The following paragraph is added to the "Analysis of Real-Time PCR Results" section on dissection, along with a reference to paragraph 43.:

"The increase expression in NAD2 in PVY-resistant Sante cultivars, which mirrors the mitochondrial ROS amplification observed in Arabidopsis rtp7 mutants during immune responses. Similar to the pathogen effector Avr-Pita in rice, which suppresses mitochondrial ROS by targeting OsCOX11, PVY may employ analogous virulence strategies to disrupt ROS-mediated defenses in susceptible cultivars. The significant increase in NAD2 expression in Sante is consistent with findings in Sclerotinia sclerotiorum infections, where mitochondrial ROS accumulation via QCR8 dysfunction promotes resistance (1). This suggests that redox hubs play a conserved role across pathosystems. Unlike studies focusing on effector-mediated ROS suppression, however, our work highlights host-driven ROS potentiation as a critical resistance mechanism, advancing the paradigm of mitochondrial redox dynamics in plant-virus interactions."

-Also we added following paragraph to compare our result, it was added in the "The miRNA Target Prediction for Hub Genes in Potato infection Response to PVY" section:

" Our discovery of multi-miRNA regulation of potato hub genes aligns with findings in tomato, where miR172 targets AP2/ERF factors to enhance late blight resistance (2). Similarly, conserved miRNAs like stu-miR5303, linked to stress responses in potato (3), highlight evolutionary conservation in Solanaceae immunity. Unlike single miRNA-target models, our network-based approach underscores combinatorial miRNA control."

The addition of these sentences was for the purpose of eliminating restrictions. This is added to the ends of the results:

" This study advances our understanding of the molecular pathways underlying PVY resistance in potatoes. However, several constraints warrant consideration. First, relying on existing RNA-seq data from various sources introduces potential confounding factors, such as discrepancies in experimental design (e.g., pathogen isolates and growth conditions) and genetic backgrounds of cultivars. While qPCR validation of pivotal genes such as NAD2 strengthened our conclusions, subsequent research should prioritize de novo transcriptomic profiling under standardized infection protocols to resolve cultivar-specific responses. Second, although greenhouse assays are valuable for providing controlled mechanistic insights, they fail to account for ecological variables prevalent in agricultural fields, such as aphid transmission efficiency, soil microbiota, and abiotic stress. To address this, we recommend multi-season field evaluations of genetically engineered lines (e.g., CRISPR-modified NAD2 variants or miRNA-overexpressing cultivars) in PVY-endemic zones to evaluate the durability of resistance and the agronomic performance. Despite these limitations, our multi-omics strategy identifies evolutionarily conserved nodes (e.g., NAD2 redox signaling and AP2/ERF transcriptional cascades) with translational potential for enhancing specific traits in potato breeding."

4- Grammar/Clarity: Fix minor errors (e.g., "lacuna persists" → "gap remains"; "mitochondrial linear membrane" → "mitochondrial inner membrane"). Simplify complex sentences (e.g., break down the 60-word abstract sentence).

Response: All requested revisions have been implemented throughout the text, and the changes are clearly marked in the manuscript using track changes.

5- Data Presentation

Strengths: Effective use of networks (Figures 2–3) and enrichment plots (Figures 4–7). Codon usage metrics (Figure 9) are novel and well-visualized.

Recommendations:

Alternative Visualizations:

a) Heatmaps: Show expression trends of hub genes across cultivars/timepoints.

b) Volcano Plots: Highlight DEGs from RNA-seq data (if available).

c) Pathway Maps: Annotate KEGG pathways (e.g., oxidative phosphorylation) with gene expression overlays.

Response: We sincerely appreciate the reviewer’s constructive suggestions regarding alternative visualization techniques. We agree that such visualizations—such as heatmaps, volcano plots, and pathway maps—are valuable tools for presenting gene expression data in more intuitive formats. However, we respectfully clarify that our study did not generate de novo transcriptomic data but instead built upon previously published RNA-seq datasets (Ross et al., 2022). The primary objective of our manuscript was to focus on downstream network-based analyses, including hub gene identification, codon usage bias, promoter motif profiling, and miRNA interactions, based on already curated sets of differentially expressed genes (DEGs).

As such, we do not have access to the raw gene expression matrices or sample-level metadata required to generate heatmaps across timepoints or cultivars, nor to construct volcano plots highlighting statistical distributions of DEGs. Similarly, due to the limited resolution of expression-level detail in the source datasets, we were unable to overlay gene expression intensity on KEGG pathway maps in a reliable and replicable manner.

Nevertheless, we have endeavored to maximize the visual clarity and biological insight of our results by providing robust enrichment plots (Figures 4–7), detailed network and subnetwork visualizations (Figures 2–3), and codon usage analysis (Figure 9), which together contextualize the regulatory and functional significance of the hub genes identified. Our integrative approach—linking PPI networks, promoter elements, codon usage, and miRNA regulation—provides a complementary perspective to standard expression-level analysis.

We hope the reviewer will understand the scope of our study and the data limitations we faced and will agree that our presentation is still rigorous and informative within this framework.

6- Figure Improvements:

• Label axes clearly in Figure 9 (e.g., "GC Content (%)").

• Include p-values or confidence intervals in Figure 10 (error bars are unclear).

Response: We edited figure 9 and 10 for clearer.

7- Tables:

• Merge Tables 2 and 3 (hub genes and clusters) to reduce redundancy.

• Add a summary table for miRNA-gene interactions (Figure 8).

Response: Table 2 specifically lists hub genes with their rankings and centrality scores, while Table 3 summarizes the functional clusters identified in the network. Since these tables serve distinct purposes—highlighting key genes vs. functional modules—we believe separating them enhances clarity. Merging them could lead to redundancy or confusion, as they address different aspects of the analysis.

Figure 8 already provides a comprehensive visualization of miRNA-gene interactions, explicitly labeling miRNAs and their target hub genes. Any table just cause redundancy we increase the quality of figure for better clear.

8- Reproducibility: Share code/scripts for bioinformatics analyses (e.g., Cytoscape workflows).

Response: We appreciate the reviewer’s emphasis on reproducibility and transparency in bioinformatics research. We would like to clarify that our analyses were conducted using publicly available tools via their graphical user interfaces rather than custom-written scripts. Specifically, we utilized:

• STRING (v10) for protein-protein interaction (PPI) data retrieval

• Cytoscape (v3.9.1) and its plugins (CytoHubba, CytoCluster) for network analysis and visualization

• MEME Suite (MEME, Tomtom, GOMo tools) for motif discovery and promoter analysis

• psRNATarget for miRNA-target prediction

• R software (for codon usage analysis), applied via basic statistical functions without scripting

As no custom scripts were used, there are no code files to share. However, we have ensured that all tool versions, parameters, and input data sources are clearly documented in the Materials and Methods section to support reproducibility.

9. Impact: Discuss translational applications (e.g., breeding targets, RNAi strategies) earlier in the discussion.

Response: We thank the reviewer for this insightful suggestion to improve the structure and impact of our Discussion. In response, we have revised the beginning of the Discussion section to introduce the translational relevance of our findings earlier, particularly highlighting how the identified hub genes (e.g., NAD2) and miRNA interactions may serve as:

• Candidate targets for molecular breeding and marker-assisted selection for PVY resistance

• RNAi or CRISPR-based engineering targets to modulate defense pathways, redox signaling, and viral fitness

• Foundations for codon deoptimization strategies to impair viral translation and replication

These applied perspectives now appear within the first two paragraphs of the revised Discussion to ensure that the potential practical value of our study is immediately visible to readers.

We believe this restructuring improves the clarity and translational relevance of our work, as the reviewer suggested.

10. Supplementary Data: Provid8- e raw qPCR Ct values and primer validation data.

Response: We appreciate the reviewer’s request for supplementary validation data to ensure the robustness of the qPCR analysis. While our qPCR instrument outputs a consolidated PDF file that contains a mix of unrelated and raw internal data not suitable for publication or extraction as standardized raw Ct tables, we have taken several steps to ensure transparency and reproducibility:

1. Figure 10 already presents the relative expression levels (fold changes) of NAD1, NAD2, and NAD3 across both cultivars, derived from Ct values using the 2^-ΔΔCT method.

2. All qPCR reactions were performed with biological and technical replicates, and melt curve analyses confirmed single specific products, ensuring amplification specificity.

Should the editorial board require a summary table of mean Ct values and standard deviations, we would be happy to prepare a simplified data table derived from our internal analysis and include it as a supplementary Excel or table file. We hope this clarification addresses the reviewer’s concerns while respecting data formatting limitations.

Reviewer: 2

1- I identified three major issues that require significant changes and improvement before the article can be accepted for publication. The article is analysing previously obtained and published data of Ross et al. (Viruses 2022, 14(3), 523; https://

---

## [Decision Letter · Decision Letter 1]

8 Jul 2025

Dear Dr. Ghorbani,

Thank you for submitting your manuscript to PLOS ONE. After careful consideration, we feel that it has merit but does not fully meet PLOS ONE’s publication criteria as it currently stands. Therefore, we invite you to submit a revised version of the manuscript that addresses the points raised during the review process.

We look forward to receiving your revised manuscript.

Kind regards,

Rajarshi Gaur

Academic Editor

PLOS ONE

Journal Requirements:

Reviewers' comments:

Reviewer's Responses to Questions

**Comments to the Author**

Reviewer #1: All comments have been addressed

Reviewer #3: All comments have been addressed

2. Is the manuscript technically sound, and do the data support the conclusions?

Reviewer #1: Yes

Reviewer #3: Yes

3. Has the statistical analysis been performed appropriately and rigorously?

Reviewer #1: Yes

Reviewer #3: Yes

4. Have the authors made all data underlying the findings in their manuscript fully available?

Reviewer #1: Yes

Reviewer #3: Yes

5. Is the manuscript presented in an intelligible fashion and written in standard English?

Reviewer #1: Yes

Reviewer #3: Yes

Reviewer #1: This article can be processed for publication. The revision has addressed all reviewer concerns appropriately and clearly.

Reviewer #3: I went through the revised manuscript entitled 'Integrated Gene Network Analysis and Experimental Validation Identify Key Hub Genes in Potato Response to Potato Virus Y Infection'. The manuscript appears far better now. The following are my suggestions.

1. Please do not start the Abstract or Introduction with The potato. Simply mention Potato

2. The references in text are somewhere names of authors and at some places numbered. Please make it uniform.

3. Italicize cis in all cis-regulatory elements

4. Please mention the source(s) of miRNAs used in psRNATarget in lines 143-147

5. Line 310 enrichment analysis

**Do you want your identity to be public for this peer review?** For information about this choice, including consent withdrawal, please see our Privacy Policy

Reviewer #1: No

Reviewer #3: **Yes: ** Kunal Mukhopadhyay

---

## [Author Response · Author response to Decision Letter 2]

8 Jul 2025

Reviewers' Comments to Author:

Reviewer: 3

1. Please do not start the Abstract or Introduction with The potato. Simply mention Potato

Response: Thank you for your comments. We removed “The”.

2. The references in text are somewhere names of authors and at some places numbered. Please make it uniform.

Response: We made all of them formatted with Plos One instructor.

3. Italicize cis in all cis-regulatory elements

Response: Done

4. Please mention the source(s) of miRNAs used in psRNATarget in lines 143-147

Response: We add a sentence to clear this section.

5. Line 310 enrichment analysis

Response: We corrected

---

## [Decision Letter · Decision Letter 2]

22 Jul 2025

Integrated Gene Network Analysis and Experimental Validation Identify Key Hub Genes in Potato Response to Potato Virus Y Infection

PONE-D-25-15213R2

Dear Dr. Ghorbani,

We’re pleased to inform you that your manuscript has been judged scientifically suitable for publication and will be formally accepted for publication once it meets all outstanding technical requirements.

Kind regards,

Rajarshi Gaur

Academic Editor

PLOS ONE

Additional Editor Comments (optional):

Reviewers' comments:

Reviewer's Responses to Questions

**Comments to the Author**

Reviewer #1: All comments have been addressed

Reviewer #3: All comments have been addressed

2. Is the manuscript technically sound, and do the data support the conclusions?

Reviewer #1: Yes

Reviewer #3: Yes

3. Has the statistical analysis been performed appropriately and rigorously?

Reviewer #1: Yes

Reviewer #3: Yes

4. Have the authors made all data underlying the findings in their manuscript fully available?

Reviewer #1: Yes

Reviewer #3: Yes

5. Is the manuscript presented in an intelligible fashion and written in standard English?

Reviewer #1: Yes

Reviewer #3: Yes

Reviewer #1: (No Response)

Reviewer #3: All the comments have been properly addressed. The manuscript may be accepted for publication in PLOS One.

**Do you want your identity to be public for this peer review?** For information about this choice, including consent withdrawal, please see our Privacy Policy

Reviewer #1: No

Reviewer #3: **Yes: ** Kunal Mukhopadhyay

---

## [Editor Report · Acceptance letter]

PONE-D-25-15213R2

PLOS ONE

Dear Dr. Ghorbani,

I'm pleased to inform you that your manuscript has been deemed suitable for publication in PLOS ONE. Congratulations! Your manuscript is now being handed over to our production team.

Kind regards,

on behalf of

Professor Rajarshi Gaur

Academic Editor

PLOS ONE